# Proposing BCG Vaccination for *Mycobacterium avium* ss. *paratuberculosis* (MAP) Associated Autoimmune Diseases

**DOI:** 10.3390/microorganisms8020212

**Published:** 2020-02-05

**Authors:** Coad Thomas Dow

**Affiliations:** McPherson Eye Research Institute, University of Wisconsin, 9431 WIMR, 1111 Highland Avenue, Madison, WI 53705, USA; ctomdow@gmail.com

**Keywords:** *Mycobacterium avium* ss. *paratuberculosis*, MAP, Bacille Calmette–Guerin, BCG, non-specific effects, zoonosis, vaccine, autoimmune, diabetes, multiple sclerosis, Johne’s, tuberculosis, non-tuberculosis mycobacteria, NTM, Buruli’s ulcer, leprosy, bladder cancer, relapsing polychondritis, molecular mimicry, heat shock protein, HSP65, Alzheimer’s, immunosenescence, Old Friends, aerobic glycolysis, Warburg effect

## Abstract

Bacille Calmette–Guerin (BCG) vaccination is widely practiced around the world to protect against the mycobacterial infection tuberculosis. BCG is also effective against the pathogenic mycobacteria that cause leprosy and Buruli’s ulcer. BCG is part of the standard of care for bladder cancer where, when given as an intravesicular irrigant, BCG acts as an immunomodulating agent and lessens the risk of recurrence. *Mycobacterium avium* ss. *paratuberculosis* (MAP) causes a fatal enteritis of ruminant animals and is the putative cause of Crohn’s disease of humans. MAP has been associated with an increasingly long list of inflammatory/autoimmune diseases: Crohn's, sarcoidosis, Blau syndrome, Hashimoto’s thyroiditis, autoimmune diabetes (T1D), multiple sclerosis (MS), rheumatoid arthritis, lupus and Parkinson’s disease. Epidemiologic evidence points to BCG providing a “heterologous” protective effect on assorted autoimmune diseases; studies using BCG vaccination for T1D and MS have shown benefit in these diseases. This article proposes that the positive response to BCG in T1D and MS is due to a mitigating action of BCG upon MAP. Other autoimmune diseases, having a concomitant genetic risk for mycobacterial infection as well as cross-reacting antibodies against mycobacterial heat shock protein 65 (HSP65), could reasonably be considered to respond to BCG vaccination. The rare autoimmune disease, relapsing polychondritis, is one such disease and is offered as an example. Recent studies suggesting a protective role for BCG in Alzheimer’s disease are also explored. BCG-induced energy shift from oxidative phosphorylation to aerobic glycolysis provides the immunomodulating boost to the immune response and also mitigates mycobacterial infection—this cellular mechanism unifies the impact of BCG on the disparate diseases of this article.

## 1. Introduction

Bacille Calmette–Guerin (BCG) vaccination was developed nearly one hundred years ago and remains the only vaccine to fight tuberculosis (TB), the result of infection by *Mycobacterium tuberculosis*. BCG is the most widely used vaccine in human history with more than four billion doses given while at the same time maintaining a strong safety record [1,2]. BCG is effective against other pathogenic mycobacteria: *Mycobacterium avium*, *Mycobacterium leprae* and *Mycobacterium ulcerans*; the infectious agents that respectively cause cervical lymphadenitis, leprosy and Buruli’s ulcer [3]. BCG use is standard-of-care treatment for non-invasive bladder cancer; BCG as a bladder irrigant promotes an immune response that lessens the recurrence of bladder cancer [4]. 

*Mycobacterium avium* ss. *paratuberculosis* (MAP) is a zoonotic agent associated with a host of inflammatory and autoimmune diseases including T1D and MS [5]. BCG has been shown to benefit both T1D and MS a result that has been termed “heterologous” effects of BCG vaccination [6]. 

This paper will review the use of BCG in TB as well as examine BCG in mycobacterial infections other than TB. It will discuss BCG use as an adjunct to bladder cancer treatment. Moreover, it will discuss the heterologous effects of BCG vaccination particularly as it relates to autoimmune diseases T1D and MS and propose that the benefit is due to MAP mitigation in these diseases. Lastly, this paper will suggest a therapeutic role for BCG vaccination in the rare autoimmune disease, relapsing polychondritis as well as explore its newfound therapeutic prospects in the very common Alzheimer’s disease.

## 2. BCG―The First Human Vaccinated

In 1931, Calmette recounts his research of three decades establishing that BCG was truly attenuated and would not back-mutate to virulence. The article conveys the context and trepidation of living with TB in the days before antibiotics, and the dangers to an infant being born into a tuberculous family. It also conveys Calmette’s trepidation in treating the first human with BCG.
“… when on July 1, 1921, Dr. Weill-Halle, who was then physician to the Infant Department of the Charite Hospital in Paris, came to consult us on a subject, which well might excite the conscientious scruples of the experimenter. He told us of a baby, born of a tuberculous mother, who had died shortly after delivery. The baby was to be brought up by a grandmother, herself tuberculous, and consequently its chances of survival were precarious. Could one risk on this child a trial of the method which, in our hands, had been constantly inoffensive for calves, monkeys, guinea-pigs and which had proved to be efficacious in preventing experimental tuberculous infection in these animals? We considered it our duty to make the trial, and the results were very fortunate, as the infant, having absorbed 6 mg. BCG in three doses per os, has developed into a perfectly normal boy, without ever having presented the slightest pathological lesion, notwithstanding constant exposure to infection during two years. When we saw that this child developed normally during the six months following the vaccination, we thought we need not wait any longer to try the method on other children.”[7]

## 3. BCG and Tuberculosis

Humans have been infected with *M. tuberculosis* (Mtb) for millennia; Mtb was discovered in 1882 by Robert Koch and is responsible for more deaths than any other human pathogen [8,9,10]. In the 1950’s large clinical trials were conducted with BCG both in England and the United States. The Medical Research Council of the United Kingdom tested a strain of BCG known as the Copenhagen strain, which was found to be highly effective against TB, whereas the Tice strain tested in the United States showed little benefit. Based upon these results, the respective public health agencies recommended routine vaccination in the UK while use in the United States was limited to high-risk groups only. The World Health Organization (WHO) followed the lead of the UK as did the majority of the world and recommended routine vaccination while the US and the Netherlands based TB control upon contact tracing and vaccinated only those at-risk [11].

These disparate results of BCG protection against TB were rationalized by two hypotheses: the differences were due to variation between BCG strains as it is recognized that different strains had different microbial properties [12]; alternatively, the US Public Health Service trial, implemented in Alabama, Georgia and Puerto Rico, was conducted in populations known to have exposure to assorted “environmental” mycobacteria. That exposure by itself could have provided some protection against TB that BCG could not greatly improve upon [13]. 

Currently, 90% of children worldwide are vaccinated with BCG with 120 million doses given annually [14]. According to the WHO an estimated one-third of the world’s population is latently infected with M. tuberculosis and in 2018 ten million people became ill and 1.5 million died from TB. This global trend will not achieve the WHO “End TB” goal of reducing clinical cases by 90% and fatalities by 95% by 2035 [15]. BCG very successfully protects children from extra-pulmonary TB, but does not reliably prevent adult pulmonary TB; though it has an excellent safety record, BCG and can cause disseminated disease (BCG-osis) in immunocompromised individuals [16]. 

## 4. BCG and Non-Tuberculous Mycobacteria 

As an attenuated live vaccine, BCG shares epitopes with mycobacteria other than tuberculosis—non-tuberculous mycobacteria (NTM)—plausibly providing cross-protection against NTM infections [3]. NTM are ubiquitous and can produce disease in susceptible individuals; notably, there has been an increase in NTM disease in developed countries that have discontinued routine BCG vaccination [17,18,19,20]. 

Cervical lymphadenitis is mostly caused by *M. avium intracellulare* complex (MAC). This NTM disease has increased significantly since the discontinuation of routine BCG vaccination in France [21], Sweden [22], the Czech Republic [23] and Finland [24].

Also caused by an NTM is leprosy (*Mycobacterium leprae*). Though present for millennia, there were more than 200,000 new leprosy cases registered in 2018 [25,26]. The protective effectiveness of BCG vaccination against *M. leprae* is well recognized [27]; vaccination with BCG decreases the risk of leprosy by 50% to 80%, and this benefit improves with BCG booster doses [28,29]. 

Buruli’s ulcer, caused by *Mycobacterium ulcerans*, is a necrotizing skin disease; behind tuberculosis and leprosy it is the third most prevalent mycobacterial infection worldwide [30]. While first described in the medical literature in 1948 in Australian patients [31], Buruli’s ulcer is primarily found in impoverished areas of Africa; in the Congo [32] and Uganda [33] and increasingly in West Africa [34,35,36]. BCG vaccination at birth protects children and adults from the serious osteomyelitis associated with Buruli’s ulcer [37]. Prospective trials with BCG have shown that vaccination confers rates of protection against Buruli’s ulcer ranging from 18% to 74%, with an overall protection rate of 47% [38,39]. 

## 5. *Mycobacterium avium* ss. *paratuberculosis*—MAP

MAP is another NTM. MAP causes a fatal infectious enteritis in ruminant animals called paratuberculosis or Johne’s disease. Johne’s disease of ruminants and Crohn’s disease of humans are increasingly regarded as the same disease: paratuberculosis [40,41,42,43]. Meta-analyses have shown that a majority of studies associating MAP with Crohn’s demonstrate MAP infection in Crohn’s patients [44,45]. Beyond Crohn’s, human diseases associated with MAP have been pursued due to the identification of genetic susceptibility risk that is shared for both the specific disease and for mycobacterial infection. Searches for polymorphisms of the *CARD15* (*NOD2*) [46,47,48], *SLC11a1* (*NRAMP1*) [49,50,51], *LRRK2* [52,53], *PTPN2/22* [54] and *VDR* [55] genes have been productive as they reveal susceptibilities for infection by mycobacteria due to impaired pathogen recognition or failure of phagosome maturation. Polymorphisms of these genes have been linked to MAP infection and concomitant diseases: Crohn’s disease [46,50], multiple sclerosis [46,56], Blau syndrome [46], autoimmune (Hashimoto’s) thyroiditis [57,58,59], Parkinson’s disease [52,60], rheumatoid arthritis [50,54,61], lupus [62] and T1D [55,63]. A majority of MAP researchers consider it a zoonotic agent [64].

The USDA (United States Department of Agriculture) has reported that the herd prevalence of MAP infection in United States dairy herds has increased from 21.6% in 1996 to 91.1% in 2007 [65]. MAP is present in pasteurized milk [66,67], infant formula made from pasteurized milk [68], surface water [69,70,71,72] and soil [69].

Specific vaccination against paratuberculosis with the live attenuated vaccine has been shown to prevent or reduce disease in ruminants but it also has severe side effects [73]. Interestingly, as BCG is safe and there has been success with BCG transformants carrying foreign antigens, the known “pathogenicity island” of MAP when added to BCG has been shown successful in an animal model [74].

## 6. BCG-Heterologous Effects

BCG vaccination in adults is beneficial for a diverse group of diseases. Research over the past 10 years has investigated the therapeutic benefits of BCG for an array of autoimmune, allergic, and induced adaptive immune responses to childhood infections [75,76,77,78,79,80,81,82,83]. The strong likelihood of an infectious environmental trigger for both T1D and MS can be seen in the low concordance rates in identical twins for these diseases: less than 40% for T1D and less than 30% for MS [84,85].

### 6.1. Non-Specific Effects of Vaccines

BCG vaccination induces two types of responses; not only the classic antigen-specific immune response leading to protection against TB and NTM, but also an adaptive “trained” immunity-based upon reprogramming of phagocytes that extends beyond protection against TB to other infections [86]. Recently, researchers Aaby and Benn reported on their forty-year campaign to demonstrate the efficacy of the four living vaccines, MV (measles vaccine), BCG, oral polio vaccine and vaccinia in reducing all-cause childhood mortality [87]. They introduced into the medical vernacular the “non-specific-effects” of vaccines (NSEs), a phenomenon linked to innate immune training [88]. 

### 6.2. BCG and Cancer 

In 1929, Pearl reviewed 1600 autopsies at Johns Hopkins Hospital; he found that lung carcinoma was less common in patients who died of pulmonary TB than in those who died of other causes, and went on to even suggest that “this formed sufficient evidence to support the treatment of cancer patients with tuberculin” [89]. Since then, many reports have been published refuting and supporting the causal relationship between TB and cancer both in animal studies and in humans. The studies included the circumstances where exposure to mycobacteria was via BCG vaccination instead of TB [4]. The use of BCG to treat stomach cancer was first reported in 1936 [90]. More recently, in 1970, Morton treated patients with melanoma with BCG intra-lesion injection and reported complete lesion regression in 684 out of 754 lesions injected, with some patients showing regression of non-injected lesions that were close to the injected lesions [91,92]. While BCG treatment of most cancers has not outperformed other therapeutic modalities, it remains the standard of care for non-invasive bladder cancer [93,94].

### 6.3. BCG and Bladder Cancer 

Morales’ study of BCG in 1976 led to BCG becoming the standard of care for non-invasive bladder cancer [95]. Approximately 80% of patients with bladder cancer are initially diagnosed with non-invasive stage (non-muscle invading) bladder cancer [96]. Recent meta-analysis confirms BCG use after bladder tumor resection decreases the risk of recurrence and progression [97]. The mechanism of action of BCG as an immunotherapeutic agent against bladder cancer involves both innate and adaptive immune responses [98,99]. A systemic immune response arises after bladder BCG therapy; this is manifest by increased lymphoproliferation, mycobacteria-specific humoral responses, conversion of the PPD skin test from negative to positive and increased serum levels of cytokines and chemokines [100]. 

### 6.4. MAP, BCG and Autoimmune Diabetes 

T1D is most often seen in childhood and in young adults and occurs with autoimmune-mediated destruction of the insulin-producing cells of the pancreas [101]. T1D is increasing in the last part of the 20th century [102]. Autoantibodies against the pancreatic enzyme glutamic acid decarboxylase (GAD) are detected in newly diagnosed children with T1D; these are thought to result from molecular mimicry in which a foreign bacterial antigen induces an immune response that cross-reacts with a similar host protein [103,104,105]. The GAD enzyme shares amino acid sequence and conformation with an immune-dominant mycobacterial protein, mycobacterial heat shock protein 65 (HSP65) [106]. All newly diagnosed T1D children in one study had an immune response against mycobacterial HSP65 [107]. It has been proposed that MAP is the source of mycobacterial HSP65 and thus an environmental trigger for T1D. A large and increasing body of work has subsequently solidified the association of MAP and T1D [83,108,109,110,111,112,113,114,115,116,117,118,119,120,121]. These studies have also confirmed genetic factors linking risk for MAP and T1D [63]. Additional MAP peptides have been identified that are homologous with pancreatic proteins [83,110,117] and the immune response to these MAP peptides cross-react with the classic diabetes islet cell antibodies [118]. 

An example of concomitant genetic risk for mycobacterial infection and T1D (and MS) is polymorphisms of the *SLC11a1* (formerly *NRAMP1*) gene. The loss of function of this gene associated with T1D and MS results in the failure of phagosome acidification when responding to mycobacteria [56,63]. 

BCG is found to have benefits for chronic T1D patients. Strikingly, BCG vaccination of long-standing T1D individuals, followed by a booster in 1 month, resulted in the control of blood sugar (seen after a delay of three years). The effect was durable with normal blood sugars eight years after the vaccination [122]. The beneficial effect is postulated to be due to a “reset” of the immune system [123]. An alternative explanation is that BCG vaccination results in mitigation against MAP [124].

### 6.5. MAP, BCG and Multiple Sclerosis

MS is a chronic, inflammatory demyelinating disease of the central nervous system clinically characterized by a broad variety of neurological signs and symptoms. Although intense research continues in the MS field, the etiology and exact pathogenic mechanisms remain poorly understood. It is, however, thought to be a multifactorial disease caused by autoimmune processes [124]. Clinical evidence suggests that an autoimmune response against myelin, stimulated by an infectious agent, contributes to this disease [125]. 

Studies determining susceptibility for mycobacterial infections have identified host genetic factors that increase risk for these infections; examples are polymorphisms in the major histocompatibility complex, TLR, vitamin D receptor genes, genes encoding IFN-gamma signaling components and *SLC11A1* [56]. As with T1D, these genes have a central role in determining risk mycobacteria infection and related autoimmune diseases, including MS [126].

There is a recognized association between MAP and MS. First studied on the island of Sardinia where MAP is endemic [127], the association is also studied in Japan, where a seroprevalence study confirmed the association between MAP and MS [128]. Supporting the association of MAP and MS is the detection of MAP DNA in the peripheral blood of MS patients [129]. As with T1D, molecular mimicry is felt to play a triggering role in MS; antibodies against MAP-specific protein MAP_2694_295–303_ and MAP pentapeptide (MAP_5p) are prevalent in the CSF and serum in MS [130]. These peptides have homology with a component of myelin, the myelin basic protein (MBP) [131]. MBP is known as a prime target of autoimmune demyelination [132]. 

First started twenty years ago, trials using BCG vaccination as an adjuvant therapy for MS patients demonstrated beneficial effects. The crossover trial revealed that a single BCG vaccination decreased magnetic resonance imaging-based disease activity in patients with MS [133]. More recent trials showed that BCG vaccination reduces the characteristic magnetic resonance imaging activity in patients with MS as well as clinically isolated syndrome (CIS) [134]. CIS is an initial clinical neurologic episode that is suggestive of MS and may later manifest as MS. 

## 7. Proposing BCG for Relapsing Polychondritis

Relapsing polychondritis (RP) is a rare but well-described autoimmune disease characterized by repeated inflammatory attacks of cartilage. The main clinical features are auricular and nasal chondritis, non-erosive seronegative arthritis, tracheobronchial inflammation, ocular inflammation, audio–vestibular disease and systemic vasculitis [78,135]. RP likely results from a combination of a genetic susceptibility, a triggering factor and a subsequent abnormal autoimmune reaction [136,137,138,139,140]. There is a concomitant genetic susceptibility between *HLA-DR4* and RP [141,142]. This same RP genetic susceptibility also confers the risk of T1D [143,144]. Moreover, *HLA*-DR polymorphisms are associated with the risk of non-tuberculous mycobacterial infection [145]. 

Antibodies from RP patients cross-react with both cartilage antigens and mycobacterial heat shock protein [146]. MAP may be the source of the mycobacterial HSP, an immunodominant protein that shares sequence and conformational elements with several human host proteins [49]. These findings in RP, not unlike autoimmune diseases T1D and MS, both showing benefit from BCG vaccination, would suggest a therapeutic consideration of BCG for RP.

## 8. BCG and Alzheimer’s Disease 

While there has been laudatory success at reducing mortality from many chronic diseases such as diabetes, HIV, heart disease, and most cancers in the past 15 years, mortality from Alzheimer’s disease (AD) has grown by more than 123% [147]. Today, 1 in 10 individuals older than 65 years is diagnosed with AD. This number doubles every 10 years until at the age of 85 years, nearly 50% will have developed the disease [148]. Clearly, this is a public health emergency [149].

Studies show that BCG vaccination has a beneficial effect on neuroinflammation in an animal model of AD [150,151]. The pathology of AD includes accumulated amyloid plaques, tau tangles and persistent inflammation. The microglia, the immune representatives in the CNS, are activated by amyloid and early in AD they clear the abnormal protein aggregates. As the disease progresses the microglia are no longer able to remove amyloid; this results in sustained inflammation along with more immune cell recruitment and pro-inflammatory cytokine production [152]. 

A recent population study found an inverse relationship between BCG vaccination and the incidence of Alzheimer’s disease. The populations studied showed a lower prevalence of AD in countries with high BCG coverage. The authors hypothesized that exposure to BCG decreases the prevalence of AD due to a modulation of the immune system. They proposed testing their hypothesis by evaluating bladder cancer patients who received BCG comparing them to bladder cancer patients for whom BCG was not part of their recommended treatment [153]. They found that bladder cancer patients treated with BCG were significantly less likely to develop AD compared to those not similarly treated. The mean age at diagnosis of bladder cancer was 68 years. AD was diagnosed at a mean age of 84 years. BCG dramatically reduced the risk of developing AD. Those treated with BCG had four-fold less risk for developing AD compared to patients not treated with BCG. The authors state that confirmation of their retrospective study would support prospective studies of BCG in AD [154].

This exciting prospect—the protective use of BCG in AD—still begs the question: what is being prevented or treated? Does BCG overcome immunosenescence associated with “the twilight of immunity” [155]? Or, with the success of BCG in treating tuberculosis, NTM infection and autoimmune diseases associated with MAP, is there a suggestion of an infectious cause for AD or specifically for a mycobacterial infection?

Infiltration of the brain by pathogens may trigger or act as co-factors for Alzheimer’s disease; Herpes simplex virus type 1 (HHV-1), *Chlamydia pneumoniae* and *Porphyromonas gingivalis* are most frequently implicated [156]. It may also be an aggregate burden of infection that leads to the pathology of AD [157]. Cytomegalovirus (CMV) also a herpes virus (HHV-5) has been associated with immunosenescence that may drive the pathology of AD [158]. *P. gingivalis*, found in the brain of AD patients eludes microglial removal via a well-described virulence factor, gingipains [159]. There is also sporadic mention of mycobacteria in a causal role for AD [160,161]. Noteworthy is the fact that an anti-mycobacterial antibiotic, rifampicin, inhibits the pathology of AD in an animal model [162]. Moreover, rifampicin was found to have preventative effects on preclinical and prodromal AD patients [163].

## 9. Discussion 

In 2014 Netea summed up current thinking about the expanded therapeutic use of BCG:
“…despite the epidemiological evidence for heterologous protective effects of BCG vaccination, the perceived lack of biological plausibility has been a major obstacle in recognizing and in investigating these effects.”[164]

The missing biological plausibility may be found in BCG’s ability to produce a cellular energy shift from mitochondrial oxidative phosphorylation to aerobic glycolysis. Aerobic glycolysis was originally found in cancer (Warburg effect) [165,166]. In response to mycobacterial infection, there is the activation of innate and adaptive immune response creating an immune remodeling that is fueled by aerobic glycolysis. The ability of pathogenic mycobacteria to subvert the host antimicrobial response may lie in its ability to interfere with the metabolic switch to aerobic glycolysis allowing mycobacterial persistence and pathogenicity [167,168]. This shift to aerobic glycolysis and its associated macrophage activation is seen in BCG’s effect in T1D [169], MS [170], AD [171] and bladder cancer [172] as well as response to mycobacterial infection [173]. The “Old Friends” theory suggests that microbial exposure at a young age assists the developmental regulation of the immune system. Consequently, rising incidences of chronic inflammatory conditions (autoimmune diseases, allergic disorders, and some types of cancer) seen in high-income countries may be attributable to dysfunctional immune regulation promoted by the hygienic lifestyle that limits contact with the Old Friends [76].

In the past ten years, there have been several clinical trials that re-introduce non-pathogenic BCG to stimulate immune remodeling against diverse infectious, autoimmune and allergic diseases [75,76,82,134]. Understanding that aerobic glycolysis, as stimulated by BCG, decreases amyloid-mediated neuronal death perhaps AD will soon be added to this list [154,169].

This article aimed to assign biological plausibility to the benefit of BCG vaccination, identifying the role of aerobic glycolysis in anti-mycobacterial prevention and treatment for MAP-associated autoimmune diseases as well as its role in immune bolstering to mitigate age-related immunosenescence. With the complex and often confounding concomitant study of infection, autoimmunity and cancer, BCG offers a parsimonious path: for those who had little exposure to the Old Friends, they may find a friend in BCG.

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
