# Peer review of "Proposing BCG Vaccination for Mycobacterium avium ss. paratuberculosis (MAP) Associated Autoimmune Diseases"

_microorganisms, 2020, doi:10.3390/microorganisms8020212_

Round 1
Reviewer 1 Report
This is a well-written, thought-provoking article that brings together many diverse lines of investigation related to mycobacteria and so called autoimmune diseases.
Only one small suggestion is offered to improve this manuscript.
Line 135-136 states: "Specific vaccination against MAP with live attenuated vaccine can prevent or reduce disease in ruminants but not without severe side effects (73)."
The cited reference does not mention much about adverse side effects. A better citation would be:
PA Windsor and J Eppleston. Lesions in sheep following administration of a vaccine of a Freund’s complete adjuvant nature used in the control of ovine paratuberculosis. New Zealand Veterinary Journal 54(5), 237-241, 2006.
Author Response
Thank you for your review and comments about the manuscript. I have read the reference that you suggested and understand that most of the side effects associated with vaccination are attributable to how the animal is restrained, correct site of injection, experience of the vaccinator and placement of the vaccine. Are you suggesting that I substitute this reference in place of the other?
thank you for the clarification.
t dow
Reviewer 2 Report
The manuscript does detail the BCG potential in the treatment of disease but really is more than a review and a proposal. The authors will need to re-write their manuscript to highlight the pieces that are important relative to what they have referenced. Minimize the "quoted" aspects since they govern almost entire pieces of subsets of the manuscript this likely would require copyright approval with the relative amount included. The authors could include tables and figures- outlining the benefit versus specific know side effects and the proposal algorithms for the vaccination process. Overall this is not a proposal- but a generalized summary with a proposed thought.
Author Response
Thank for your review of the manuscript. The manuscript is labeled a "concept paper" and concern over methods and design are not applicable. Indeed, you have identified this with your last remark:
Overall this is not a proposal- but a generalized summary with a proposed thought.
As BCG is already used for T1D and MS, the concept proposed is that these diseases be recognized as associated with MAP.
With regard to the quotations, it has been amazing to me after a 40 year career in clinical medicine and research that, for the most part, vaccination with BCG was only a consideration for missionaries or equally at-risk individuals. Very little information was disseminated in the US about the fact that 120 million doses continue to be given annually. Now I am proposing millions more be vaccinated. The fascinating retelling, by Dr. Calmette, of the first patient vaccinated I feel is an integral part of this story. The second of the quotations basically says that in 2014, because the method of action of BCG was undetermined, its use was restrained. The manuscript details the method of action in the hopes that this no longer presents a hurdle for investigation of BCG in new diseases.
If the editors decide that the length of the quotes present a copyright risk, I will certainly agree to remove them; otherwise I feel they bookend the start of BCG as well as a new starting point for its use.
Respectfully.
Tom Dow